# Characterization of a Droplet Containing the Clustered Magnetic Beads Manipulation by Magnetically Actuated Chips

**DOI:** 10.3390/mi13101622

**Published:** 2022-09-28

**Authors:** Sheng-Huang Yen, Pei-Chieh Chin, Jun-Yu Hsu, Jr-Lung Lin

**Affiliations:** Department of Mechanical and Automation Engineering, I-Shou University, Kaohsiung 84001, Taiwan

**Keywords:** magnetically actuated, droplet, point-of-care testing, digital microfluidics

## Abstract

A magnetically actuated chip was successfully developed in this study to perform the purpose of transportation for a droplet containing clustered magnetic beads. The magnetic field gradient is generated by the chip of the two-layer 4 × 4 array micro-coils, which was commercially fabricated by printing circuit board (PCB) technology. A numerical model was first established to investigate the magnetic field and thermal field for such a micro-coil. Consequently, the numerical simulations were in reasonable agreement with the experimental results. Moreover, a theoretical analysis was derived to predict the dynamic behaviors of the droplets. This analysis will offer the optimal operation for such a magnetically actuated chip. This study aims to successfully implement the concept of “digital microfluidics” in “point-of-care testing” (POCT). In the future, the micro-coil chip will be of substantial benefit to genetic analysis and infectious disease detection.

## 1. Introduction

In past decades, micro-electro-mechanical systems technologies have contributed to the wide and rapid development of microfluidic devices with many different functions. The functions of transporting, mixing, separating, detecting, and sensing are exhibited in micro-total analysis systems (µTAS). In implementing the aforementioned functions, transporting the sample and reagent plays an imperative role, which is required in the applications of medicine, biology, and the environment. Two microfluidic technologies, namely continuous flow or droplet-driven schemes, have successfully demonstrated miniaturized μTAS for transportation [1]. The aforementioned approaches are promising, but major challenges still exist in real-world biosamples with microfluidic devices and their reliance on equipment for actuation [2]. In such tasks, droplet-based platforms were used for the storage and processing of samples and reagents demonstrating significant potential without the aid of micropumps [1]. Droplet-actuated microfluidic devices have made substantial progress toward integrating essential manipulation for bio-analysis.

In a continuous flow scheme, transport is facilitated by the devices of micropumps. Briefly, micropumps can be divided into either non-mechanical or mechanical depending on whether the elements are fixed or movable [3]. Non-mechanical pumps, which have fixed elements, must import external energies to convert momentum to facilitate fluid flow. Converted methods include electro-hydrodynamic, magneto-hydrodynamic, electro-osmotic, electrowetting, bubble type, and electrochemical approaches [3]. Conversely, mechanical pumps must integrate movable elements (i.e., actuators) into a body chamber. To drive the fluid flow, external energies must also be imported into the actuators. The driven modes of actuators include electrical, electrostatic, pneumatic, piezoelectric, and magnetic types [3]. However, valves must be used in mechanical micropumps to restrict the flow direction [4]. These valves can generally be categorized as check valves [5], flapping valves [6,7], or peristaltic valves [8,9]. In achieving valveless pumping, mechanical micropumps have been successfully developed with simple geometries [10]. However, these valves with mechanical micropumps can damage and clog the cells or particles in the process of fluid flow [11]. Additionally, these valves continuously consume reagents and biosamples during transportation. Furthermore, these valves are constantly applied under such a high pressure; thus, the valve materials should overcome fatigue damage for long-term operation [12].

Droplet-based microfluidics, also named digital microfluidics, have recently gradually gained a considerable amount of attention. One major advantage of droplet-based microfluidic devices lies in their manipulation of droplets on an open surface. In addition, droplets are not only self-contained systems but also function as reaction chambers and transportation units. The droplet-actuated platforms are particularly useful for point-of-care testing because of their simplicity, portability, and ability to store samples and reagents on the chip. Many actuated methods have been successfully proposed to manipulate the movement of the droplets, including electrowetting on dielectric [13], thermocapillary force [14], surface acoustic wave [15], electrophoresis [16], optical force [17], and magnetic force [18,19,20,21,22]. Among all the actuation methods, magnetic-actuated droplet manipulation [18,19,20,21,22] demonstrates some advantages because of its flexibility, long distance, large driving force, and easy operation.

Magnetic beads not only serve as an actuated medium but are also carriers for biomolecules; for example, super-paramagnetic particles are suitable for nucleic acid binding and transfer [2]. Magnetic beads are also attractive from a practical perspective because they can be used to capture proteins or nucleic acids [23]. The extraction from crude biosamples uses various commercially available reagents and surface-coated particles to perform on-chip DNA extraction and purification [18,19,20,22]. Permanent magnets [19,20] or micro-coils [18,21,22] have been employed to transport [19,20,22], split [18,20,22], and merge droplets [19] for genetic assay. Nonetheless, the numerical simulation remains inadequate for analyzing the magnetic and thermal fields of such planar coils. If numerical simulation can be utilized to analyze the distribution of magnetic and temperature fields for the planar micro-coils, then this simulation can provide an optimal design for these micro-coils. Therefore, the time and cost of trial-and-error methods can be significantly reduced.

Therefore, a magnetic-actuated chip for transporting droplets that utilizes two-layer 4 × 4 array micro-coils was designed in this study by fabricating the PCB technology. Numerical simulation was used to investigate the magnetic and thermal fields for a micro-coil. Moreover, theoretical analysis was derived to predict the moving speed of a droplet considering the parameters, including the loading mass of magnetic beads, droplet volumes, and applied DC currents. Finally, fully automated manipulation of a droplet on an open surface was demonstrated for conducting sample transportation.

## 2. Theoretical Analysis

In a previous study, Long et al. [19] derived a theoretical analysis to categorize an “operating diagram” that describes the following three droplet regions: stable motion, breakage, and magnet release. They have derived various formulas to analyze the three different regions. A theoretical model is developed in this study to characterize the dynamic behaviors of the droplet containing clustered magnetic beads. The kinematics of the droplet is determined by the Newtonian law acting on the magnetic force (*F_B_*), Stoke force (*F_St_*), and the surface tension force (*F_γ_*). Magnetic force is expressed as follows [24]:(1)$$F_{B}=\frac{V_{b}\chi (B\cdot \nabla )B}{\mu_{o}}$$
where *V_b_* is the volume of magnetic beads, *χ* is the magnetic susceptibility of the beads, *μ**_o_* is the permeability of free space (4π × 10^−7^ H/m), and B is the magnetic flux density (i.e., magnetic field).

In principle, the droplet is driven to move by the *x*-axis magnetic force (*F_B,x_*), which is simplified by [24]:(2)$$F_{B,x}=\frac{m_{b}\;\chi \;B_{m}B_{z,Max}}{\mu_{o}\rho_{b}l_{x}}$$
where *m_b_* and *ρ_b_* are the mass and density of the clustered magnetic beads, respectively; *l_x_* is the axial distance from the center to the rim of a micro-coil; and *B_m_* is the *z*-axial magnetic field generated by the Helmholtz coil. Notably, the maximum z-direction magnetic field, that is, *B_z,Max_*, is induced by the micro-coil applied in a given DC current.

The motion of a droplet surrounded by silicon oil follows Stoke’s rule. The Stoke’s force (*F_St_*) is obtained as follows:(3)$$F_{St}=3\pi \eta_{O}d_{D}u_{D}$$
where *η_O_* is medium (oil-phase) viscosity, *d_D_* is the diameter of the droplet, and $u_{D}$ is the moving speed of a droplet. The frictional force (*F_μ_*) between a liquid droplet and a substrate surface can be expressed as the following:(4)$$F_{\mu }=\mu_{k}\Delta \rho V_{D}g$$
where *μ_k_* is the friction coefficient of the substrate surface. Δ*ρ* is the difference density between liquid droplets and the surrounding medium. *g* is the gravitational acceleration. Upon moving onto a substrate, a droplet’s surface tension is expressed in the following manner:(5)$$F_{\gamma }=\pi d_{D}\gamma_{O-W}\cos\theta$$
where *θ* is the contact angle. *γ_O_*_-*W*_ is the relative surface tension between liquid water and oil medium. In order to present the ratio between frictional force and surface tension force, the coefficient of *ζ* is introduced to define as follows:(6)$$\zeta =\frac{F_{\mu }}{F_{\gamma }}=\frac{\mu_{k}\Delta \rho V_{D}g}{\pi d_{D}\gamma_{O-W}\cos\theta }$$

By simplifying Equation (6), the coefficient of *ζ* is obtained as follows (see Appendix A):(7)$$\zeta ={\left(\frac{2(1-\cos\theta )}{3\pi^{2}\sin^{3}(2\theta )}\right)}^{1/3}\frac{\mu_{k}\Delta \rho {\left(V_{D}\right)}^{2/3}g}{\gamma_{O-W}}$$

A droplet moving speed (*u_D_*) can be determined by the force balance between the magnetic force, frictional force, and Stoke’s drag force, as shown below:(8)$$\rho_{D}V_{D}\frac{du_{D}}{dt}=-3\pi \eta_{O}d_{D}u_{D}+\frac{m_{b}\chi B_{m}^{}B_{z,Max}}{\mu_{o}l_{x}}-\mu_{k}\Delta \rho V_{D}g$$

When a droplet moves, its shape is assumed to remain unchanged. The surface tension of a droplet will not affect the droplet’s motion since it is an internal force. Additionally, when the droplet moves, determining the frictional force can be a challenge. Therefore, the surface tension force is used instead. Equation (8) can be rewritten as follows:(9)$$\rho_{D}V_{D}\frac{du_{D}}{dt}=-3\pi \eta_{O}d_{D}u_{D}+\frac{m_{b}\chi B_{m}^{}B_{z,Max}}{\mu_{o}l_{x}}-\zeta \pi d_{D}\gamma_{O-W}\cos\theta$$

The moving speed of the droplet (*u_D_*) can be solved as follows:(10)$$u_{D}(t)=\left(\frac{m_{b}\chi B_{m}^{}B_{z,Max}}{\rho_{D}V_{D}\rho_{b}\mu_{O}l_{x}}-\frac{6\zeta \gamma_{O-W}\cos\theta }{\rho_{D}d_{D}^{2}}\right)\left(1-e^{-t/\tau_{s}}\right)$$
where *τ_S_* is Stoke’s time, which is defined as $\tau_{S}=\rho d_{D}^{3}/18\eta_{O}.$ As *t* approaches infinity or is substantially larger than Stoke’s time and *U_D_* is called the terminal velocity. *U_D_* can be obtained by:(11)$$U_{D}=\left(\frac{m_{b}}{\rho_{D}V_{D}\rho_{b}}\frac{\chi B_{m}B_{z,Max}}{\mu_{O}l_{x}}-{\left(\frac{36\cos\theta }{\pi V_{D}^{2}}\right)}^{1/3}\frac{\zeta \gamma_{O-W}}{\rho_{D}}\right)$$

### 2.1. Design and Fabrication

A droplet containing the clustered magnetic beads manipulated by the two-layer micro-coils is schematically shown in Figure 1. The Helmholtz coil is used to produce a uniform transversal magnetic field (*B_m_*). The two-layer micro-coils are utilized to generate the magnetic field gradient (▽*B*). The magnetic force is linearly proportional to *B_m_* (▽*B*), which actuates the movement of the droplet. Initially, the droplet containing the clustered magnetic beads is fixed at the center of micro-coil #1 while coil #1 is turned on. Afterward, coil #1 is turned off while coil #2 is turned on. The droplet is then driven to move at the center of coil #2. The magnetic field is produced by the sequential turning on/off of the micro-coil at a constantly driving frequency. Thus, the droplet is driven to achieve transportation performance.

The photograph of the magnetic-actuated chip is displayed in Figure 2a. The proposed chip of the two-layer 4 × 4 array micro-coils was printed by PCB technology as shown in Figure 2b. Each layer is separated into 100 μm thickness. Micro-coils of the upper and lower layers partially overlap with a distance of 3.0 mm between them. Micro-coils consist of eight windings in a square shape. These micro-coils are designed to be 150 μm in width and gap. In order to generate hydrophobic properties, Teflon coatings are applied. The magnetic-actuated chip was subsequently dip-coated in a solution of 1% *w*/*w* Teflon AF 1600 (DuPont Corp., Wilmington, DE, USA) mixed with FC-40 solvent (3M Company, St. Paul City, MN, USA) and was then baked overnight at 80 °C.

### 2.2. Experiments

The experimental setup can be seen in Figure 3. The Helmholtz coil is placed on the top and bottom of the magnetically actuated chip with a given gap. This generates a uniform transversal magnetic field measuring approximately 50.0 mT at a 50 × 50 mm^2^ area. The magnetically actuated chip was placed on a platform for observation of the droplet’s kinetics. A power supply was employed to apply the DC currents on the two-layer 4 × 4 array micro-coil chip. A droplet was then dropped on the top surface of the chip. Afterward, 2.88 μm magnetic particles (MF-DEX-3000, MagQu LLC, Surprise City, AZ, USA) were dispensed into the water droplet. The magnetic beads are formed by the core of a single crystal Fe_3_O_4_ metallic sphere coated with dextran. The induced magnitude and direction of the magnetic fields were determined by adjusting the driving DC currents, while the direction of the magnetic fields produced by each coil was controlled by a custom-designed analog circuit. Images of the droplet movement were captured using a digital camera attached to a charge-coupled device (CCD, Cool SNAP HQ2, Photometrics, Huntington Beach, CA, USA). Customized software edited in LabVIEW (National Instruments, Austin, TX, USA) was utilized to control the magnitude, duration, and direction of DC currents sequentially. Notably, a thermoelectric cooler integrated with a K-type thermocouple was equipped underneath the PCB as a cooling system. The cooling temperature was controlled by utilizing a commercial PID controller to dissipate the effects of Joule heating.

## 3. Results and Discussion

### 3.1. Magnetic Characterizations

A numerical simulation was performed in this study to investigate a planar micro-coil considering magnetic and thermal fields. These fields were numerically simulated using commercial software (CFD-ACE+, CFD-RC, Huntsville, AL, USA). The modules of thermal, electric, and magnetic fields were selected to study the magnetic and thermal phenomena [25]. Heat generation and DC currents were employed on an 8-winding coil. The heat transfer of the proposed system is mainly governed by the mechanisms of heat conduction coupling with free convection. In this simulation, a 3D geometric domain was discretely created with structured hexahedral meshes. A residual criterion of 10^−8^ was employed to guarantee the convergence of each iterative solution step in the simulated process. The physical properties of the materials, i.e., copper, Teflon, and air, are listed in Table 1.

Figure 4a shows the simulated geometries of two micro-coils. The dimensions of an 8-winding micro-coil are listed in Table 2. Two different directions of DCs were separately applied to each of the coil centers. The 2D and 3D contours of the magnetic field were numerically demonstrated as shown in Figure 4b,c, respectively. Considering the application of a DC of 1.0 A in different directions, the 2D or 3D profiles of the *z*-axis magnetic field were symmetrical to the wave-shaped surfaces. The maximum value was numerically obtained as 2.84 mT at a DC of 1.0 A.

The numerical simulation results of different DC currents revealed that the 2D and 3D contours of the magnetic field were remarkably similar but demonstrated differences in the maximum values. Therefore, dimensionless parameters were introduced and analyzed in this study. Herein, the dimensionless parameters are defined as follows: *B***_Z_* = *B_Z_***/***B**_Z_*_,*Max*_ for the *z*-axial magnetic field, *F***_B_*_,_*_X_* = *F_B_*_,_*_X_***/***F_B_*_,_*_X_*_,*Max*_ for the *x*-axial magnetic force, and *x** = 2*x***/***W* for the *x*-axial distance. *B**_Z_**_,Max_* is the *z*-axial maximum magnetic field, *W* is the width of a micro-coil, and *F**_B_*_,*X*__,*Max*_ is the *x*-axial maximum magnetic force. The profile of *B***z* varied along with *x** as shown in Figure 5a. The maximum value is located near the center but not at the initial point of a coil. The value of *B***_Z_* gradually decreased with increasing *x**. The zigzag-shaped profile was observed at the upper layer of the coil. However, the profile demonstrated smoothness at the lower layer of the coil. The difference in *B***z* at the lower layer is approximately 0.834 times that of the upper layer. The profiles of *F***_B_*,*_X_* were numerically varied along with the *x** for the two different layers, as shown in Figure 6b. However, the profiles for different layers are substantially different. Notably, *F***_B_*_,*X*__,*Max*_ at the lower layer is approximately 0.3 times that of the upper layer.

The numerical simulations of the *B*_*z*,*Max*_ were then compared with the experimental measurements using a Tesla meter (TM-401, KANETEC, Tokyo, Japan) versus different applied currents, as shown in Figure 6a. The simulation and experiment results of *Bz*,*_Max_* quasi-linearly varied with the applied DC currents for the planar coil. Additionally, the numerical simulations are in reasonable agreement with the experimental measurements. The deviation between the numerical calculations and experimental measurements was less than 18.5%. Similarly, the *F_B_*_,_*x*_,*Max*_ was then calculated as displayed in Figure 6b. The *F_B_*_,_*x*_,*Max*_ results also presented linear proportions with the different DC currents.

### 3.2. Thermal Characterizations

The temperature effect is a crucial issue for magnetic actuation. Elevated temperature cannot be immediately reduced, thus decreasing the efficiency of the magnetic manipulation or damaging biosamples. The 3D steady-state heat conduction equation can be expressed as:(12)$$\nabla \cdot \left(k\nabla T\right)+\dot{q}=0$$
where *k* is the thermal conductivity and $\dot{q}$ is the heat generation density. The heat generation density ($\dot{q)}$ due to Joule heating is obtained as follows:(13)$$\dot{q}=\frac{\dot{Q}}{V_{C}}=\frac{I^{2}R}{V_{C}}=\frac{L_{C}\times I^{2}}{\sigma V_{C}(A_{C})}$$
where *I*, *σ*, *A_C_* and *V**_C_* are electric current, electric conductivity, cross-sectional area, and occupied volume of a micro-coil, respectively. The surrounding surfaces were set under an adiabatic condition. The top and bottom surfaces were dissipated by heat convection are expressed as follows:(14)$$-k\nabla T=h(T_{S}-T_{\infty })$$
where *h* is the heat convection coefficient. *T_S_* and *T*_∞_ represent the surface and circumstance temperature, respectively. In particular, the free convection coefficient (*h*) is a function of surface geometry, fluid motion, and fluid properties. Therefore, the value of h was set to 7.5 Wm^−2^K^−1^ [26] to evaluate heat generation coupled with free convection [26]. The 2D and 3D numerical contours of temperature distribution are shown in Figure 7a,b, respectively. The maximum temperature reached is 68.9 °C. Thus, the heat source generated by the 8-winding micro-coil dissipated and uniformly distributed on the chip. A dimensionless temperature is defined as *T** = *T*/*T_Max_*. The profile of the dimensionless temperature is also shown in Figure 8a. The temperature displayed a zigzag shape due to the windings of a micro-coil. However, these differences in the temperature between the 8-winding of a coil and the gaps are substantially small.

The temperature of the planar coils was experimentally measured by a top-attached K-type thermocouple. The numerical simulation and measurement comparison of the maximum temperature against the applied current are shown in Figure 8b. The figure reveals that the maximum temperatures are quadratically increased with the rise in applied currents. The numerical simulation could achieve consistency with the experimental measurements. Furthermore, the deviation between the numerical simulations and experimental measurements was less than 5.9%. This finding proved that the numerical model of the temperature field can be applied to optimal designs (such as micro-coils) in the future.

### 3.3. Transportation Characterizations

A droplet containing the clustered magnetic beads, which is located at the center of an upper-layer micro-coil as indicated in Figure 9a. Afterward, a constant electric current was applied at the lower layer of a micro-coil to generate the magnetic field. The clustered magnetic beads were then induced to move the front rim of a droplet as shown in Figure 9b, thus facilitating the movement of the droplet to the center of a micro-coil (Figure 9c). The moving time is experimentally measured in the movement process. The motion distance, that is, half a width of the micro-coil (*l_x_* = 3.0 mm), was then divided by the moving time to convert the moving speed of a droplet.

Equation (11) is used to predict the moving speed of a droplet containing clustered magnetic beads. The parameters included the loading mass of the clustered magnetic beads, droplet volumes, and applied DC currents. The electric properties of materials are listed in Table 3. Figure 10 shows the moving speed of a droplet considering the mass of magnetic beads at a constant droplet volume of 12 μL and an applied current of 0.4 A. This figure indicates that the loading mass of magnetic beads must exceed “the threshold mass” to actuate the droplets for the given droplet volume. In principle, the “loading mass” is too small (i.e., *m_b_* < 60 μg) to actuate the droplet movement. Moreover, the droplet moving speed linearly increased with the rising loading mass of magnetic beads for the theoretical analysis and experimental data.

The mass of the clustered magnetic beads is maintained at *m_b_* = 300 μg with the application of two different DC currents and the current magnitudes are 0.4 and 1.0 A. The volumes of a droplet were pipetted to be 3.0, 6.0, 9.0, 12.0, and 15.0 μL. The experimental results revealed that the movement speeds are inversely proportional to the droplet volumes for a constant DC. As expected, a small droplet volume and high DC can achieve a fast moving speed. On the contrary, a large droplet volume and low DC could lead to a slow movement speed. The theoretical analysis of moving speed, that is, the formula of Equation (11), was also compared with the experimental measurements as shown in Figure 11. The prediction results are in reasonable agreement with the measurements at a low DC of 0.4 A. However, the prediction results are substantially higher than the experiments at a high DC of 1.0 A. This phenomenon is due to the thermal effect, which was dissipated at a high electric current. Therefore, the theoretical analysis of a droplet moving speed can be used for predictions at a DC of 0.4 A while the thermal dissipation is neglected.

Finally, a liquid droplet was tested to facilitate transportation on an open surface by the proposed micro-coil chip. To reduce movement resistance, silicon oil was uniformly distributed on the PCB surface. The operating condition is as follows: *m_b_* = 300 ug, *V_D_* = 12.0 uL, and *I* = 1.0 A. Every adjacent micro-coil turns on/off by switching a frequency of 0.5 s. Figure 12 shows a series of photographs to perform the transportation of a droplet. A droplet containing the clustered magnetic beads was smoothly and continuously transported by a two-layer 4 × 4 array micro-coil chip. These results verified that the digital microfluidic chip could be used for transportation applications. In the current prototype, magnetically actuated chips are capable of transporting droplets either vertically or horizontally. The proposed system, however, does not implement curved trajectory functions. The sequence of a full cycle is displayed in Appendix A.

## 4. Conclusions

The major contribution of the droplet-based manipulation lies in its flexibility because this magnetic actuation can simultaneously handle a diverse range of biosamples and reagents. It is also easily integrated with thermal control and optical detection. Overall, the *z*-axial maximum magnetic fields linearly vary with the DC currents with the function of 2.86 × *I*. The temperature differences also vary quadratically proportional to 45.5 × *I*^2^. The numerical simulations were compared with the experimental measurements of magnetic and temperature fields, with a deviation of less than 18.5% and 5.9%, respectively. Thus, the numerical model could be a powerful tool for the optimal design of the micro-coils. Meanwhile, the proposed theoretical analysis for the dynamics of the droplet could be used to predict the motion behavior. The developed platform is expected to deliver nucleic-acid-based diagnostic assays to point-of-care testing (POCT).

## Figures and Tables

**Figure 1 micromachines-13-01622-f001:**
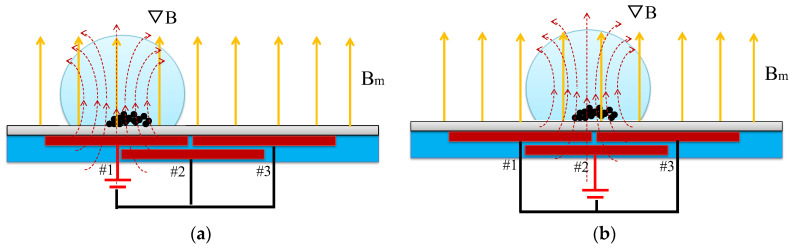
Schematic of the working principle of the two-layer micro-coil chip magnetically actuated droplet movement. (**a**) Fixed at the center of coil #1 while coil #1 is turned on. (**b**) Moved to the center of coil #2 while coil #2 is turned on and coil #1 is turned off.

**Figure 2 micromachines-13-01622-f002:**
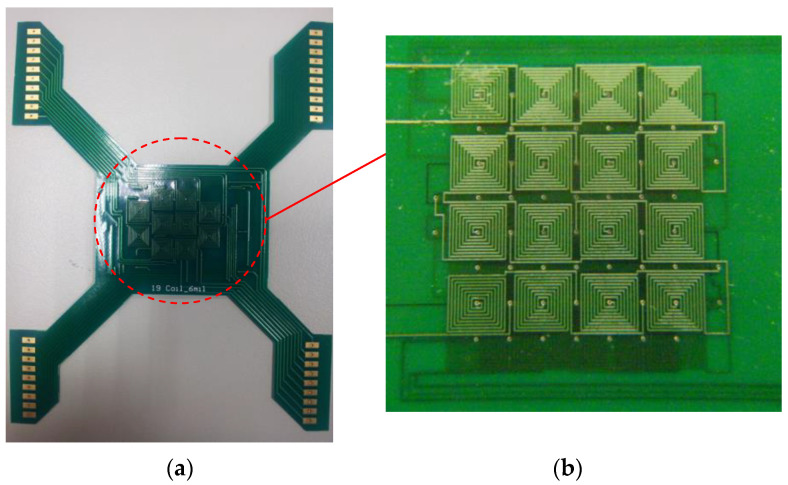
(**a**) Photograph of the magnetic-actuated chip. (**b**) Two-layer 4 × 4 array micro-coils of the PCB layout on the upper and bottom layers.

**Figure 3 micromachines-13-01622-f003:**
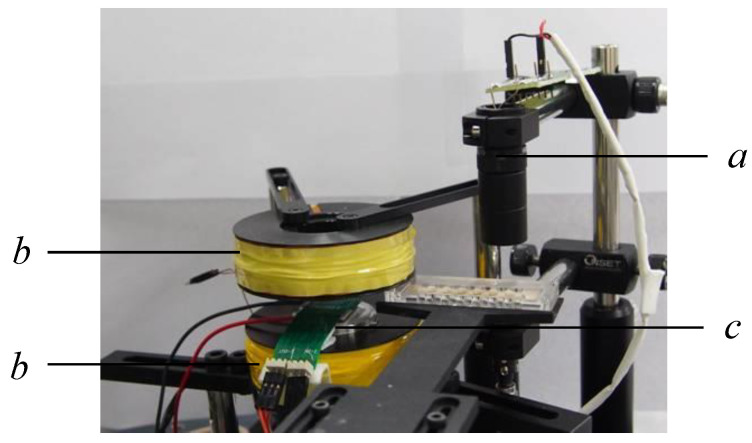
Experimental setup of the magnetically actuated platform. a: digital camera attached with a CCD; b: Helmholtz coil; c: magnetically actuated chip.

**Figure 4 micromachines-13-01622-f004:**
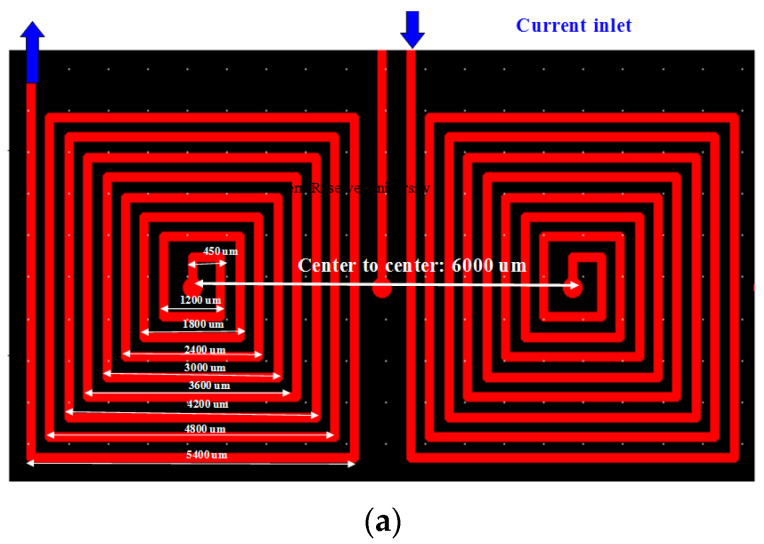
(**a**) Numerical geometry of two micro-coils. (**b**) 2D and (**c**) 3D numerical contours of the *z*-axial magnetic field at an applied electric current of 1.0 A.

**Figure 5 micromachines-13-01622-f005:**
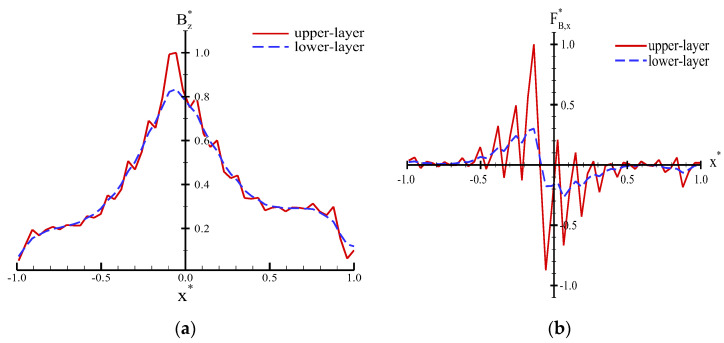
Numerical profiles of the dimensionless (**a**) *B***_Z_* and (**b**) *F***_B_*_,_*_X_* varied along with *x**.

**Figure 6 micromachines-13-01622-f006:**
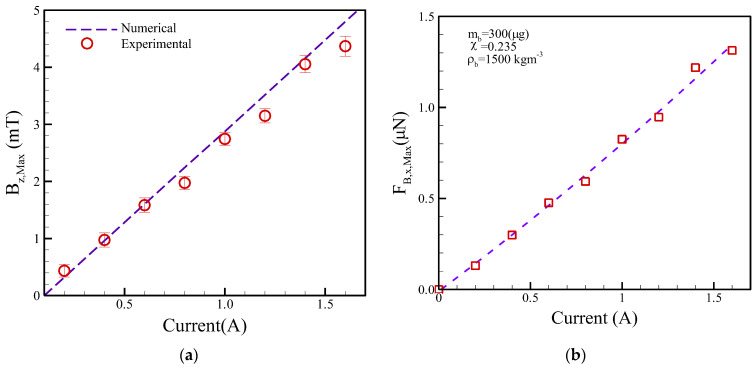
(**a**) Numerical and experimental results of *B**_Z_*_,*Max*_ and (**b**) calculations of *F_B_*_,_*_X_*_,*Max*_ versus different DC currents.

**Figure 7 micromachines-13-01622-f007:**
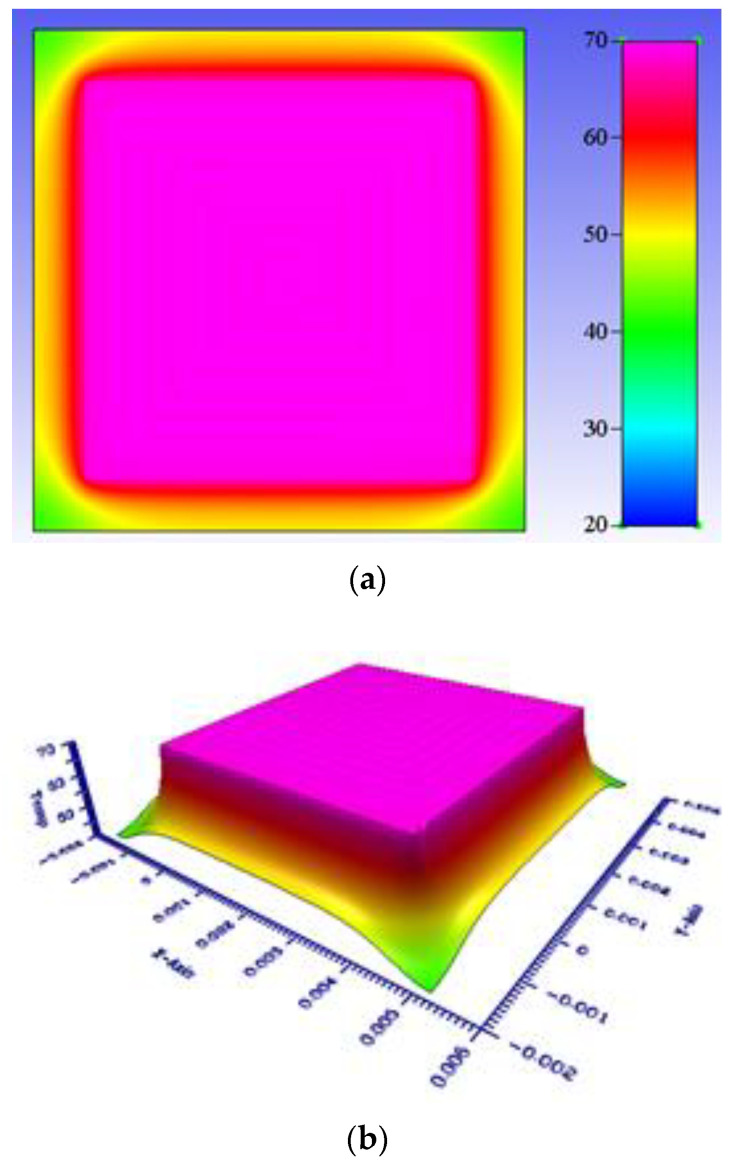
(**a**) 2D and (**b**) 3D numerical contours of the temperature field at an applied electric current of 1.0 A.

**Figure 8 micromachines-13-01622-f008:**
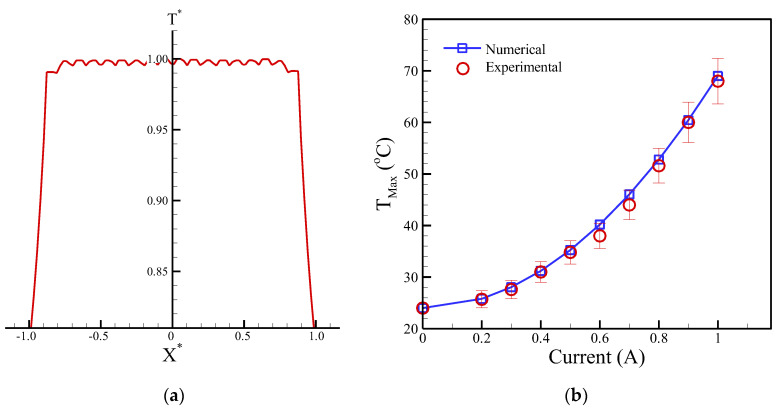
(**a**) Numerical profiles of *T** varied along with the dimensionless distance of *x**. (**b**) Numerical and experimental comparison of maximum temperature (*T_Max_*) versus different applied currents.

**Figure 9 micromachines-13-01622-f009:**
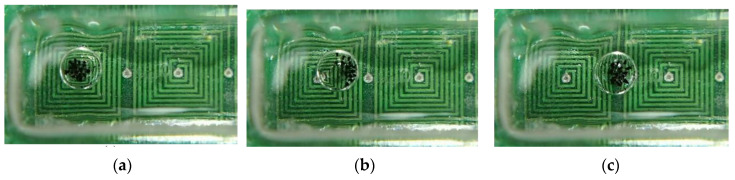
**A** series of photographs depicting a droplet actuating from (**a**) intial postion, (**b**) movement, to (**c**) final destination.

**Figure 10 micromachines-13-01622-f010:**
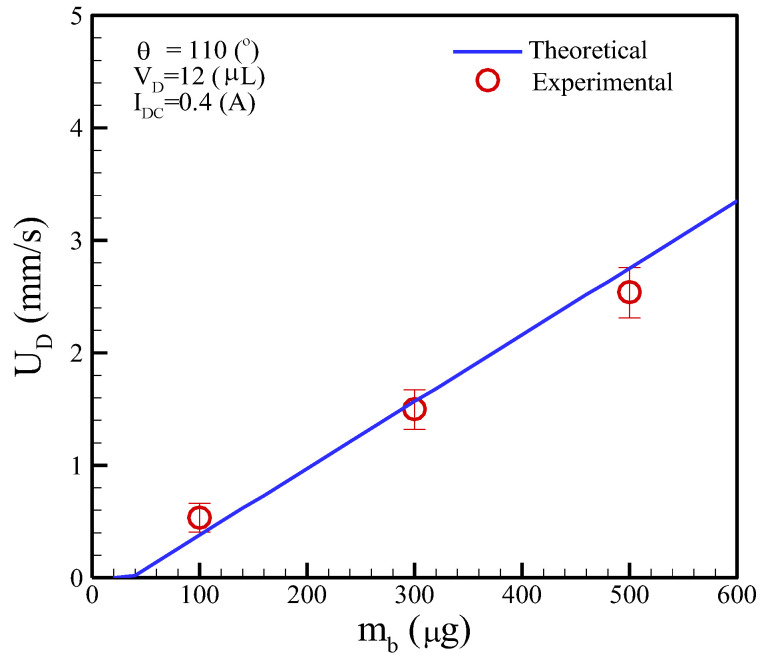
Moving speed of a droplet versus the loading mass of the magnetic beads, where *ζ* = 0.001 was chosen in this case.

**Figure 11 micromachines-13-01622-f011:**
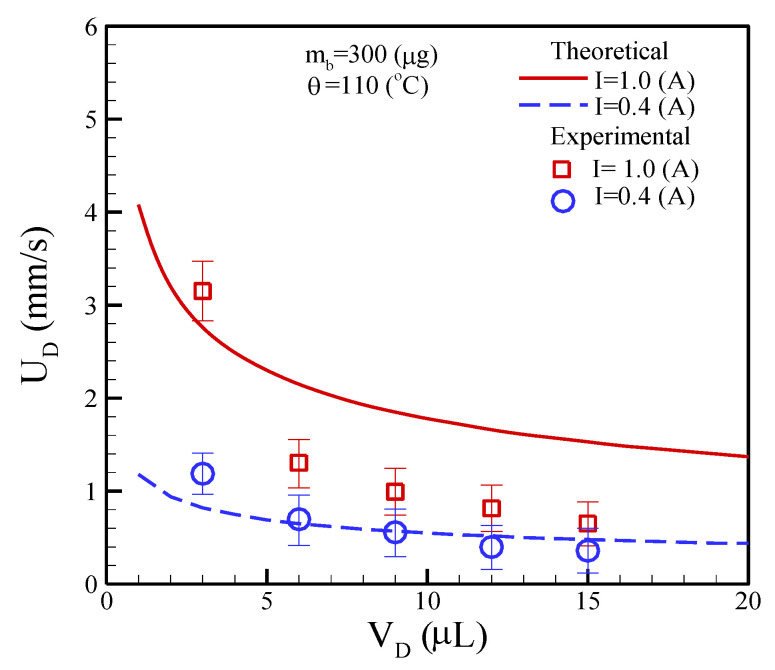
Theoretical and experimental comparisons of a droplet at different volumes versus moving speeds.

**Figure 12 micromachines-13-01622-f012:**
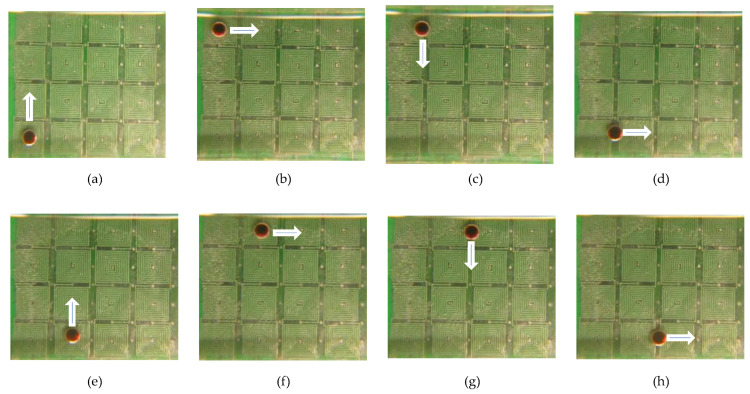
Water droplet moves along the two-layer 4 × 4 array micro-coils chip at different times. (**a**) t = 0; (**b**) t = 3.0 s; (**c**) t = 3.5 s; (**d**) t = 6.5 s; (**e**) t = 7.0 s; (**f**) t = 10.0 s; (**g**) t = 10.5 s; (**h**) t = 13.5 s; (**i**) t = 14.0 s; (**j**) t = 17.0 s; (**k**) t = 17.5 s; and (**l**) t = 20.5 s. The white arrows indicate the motion direction.

**Table 1 micromachines-13-01622-t001:** Properties of materials.

Property	Copper	Teflon	Air
density (kgm^−3^)	8960	940	1.23
electric conductivity (Sm^−1^)	5.996 × 10^7^	1.0 × 10^−4^	10^−9^
thermal conductivity (Wm^−1^K^−1^)	386	0.3	0.025
specific heat capacity (Jkg^−1^K^−1^)	385	1500	1006
convection coefficient (Wm^−2^)	none	none	7.5
relative permeability	1	1	1

**Table 2 micromachines-13-01622-t002:** Micro-coil geometry.

Geometry	Value
width (μm)	150
height (μm)	35
gap (μm)	150
winding	8
length (mm)	17.4

**Table 3 micromachines-13-01622-t003:** Properties of materials.

Property	Magnetic Bead [27]	Silicon Oil [28]	Water
density (kgm^−3^)	1500	970	1000
viscosity (Pa-s)	none	0.0149	0.001
magnetic susceptibility	0.235	none	none
surface tension (Nm^−1^)	none	0.021	0.072

## Data Availability

The data presented in this study are available within the article. The other data that support the findings of this study are available upon request from the corresponding author.

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
