# Peer review of "Characterization of a Droplet Containing the Clustered Magnetic Beads Manipulation by Magnetically Actuated Chips"

_micromachines, 2022, doi:10.3390/mi13101622_

Round 1
Reviewer 1 Report
The authors of this paper created a magnetically actuated digital microfluidics chip comprised of two-layer 4 4 array micro-coils based on printing circuit board (PCB) technology. Theoretical predictions on dynamic droplet behaviors were compared to experimental results under various operating conditions. The article is well-organized and written, and the results are clear. Furthermore, the collected data is properly analyzed and presented. It is also significant both academically and practically, and thus may be of interest to many potential Journal readers. As a result, I believe this paper can be published with minor revisions.
1. Equation 5 on Page 3: Please describe the physical meaning of a coefficient of ζ to help readers understand Equation 5.
2. Pages 11-12: Experimentally and theoretically, the authors established the relationship between droplet movement speed and droplet volume. The influence of gravity is not accounted for in the theoretical model, but I wonder if there is a gravitational effect when a droplet's size exceeds the capillary length.
3. Some researchers demonstrated that introducing a chemically modified surface energy trap can facilitate magnetic bead extraction from magnetic droplets (e.g., Park et al 2021 Jpn. J. Appl. Phys. 60 076504). It would be beneficial to include such a system in your research in the near future.
4. It is advised that this paper be thoroughly proofread before publication.
Reviewer 2 Report
In this work the authors report a magnetically actuated chip that can be used for the manipulation of droplets containing magnetic microbeads. The characterize the chip by studying the magnetic and temperature profiles caused by the arrays of Helmholtz coils and show reasonable agreement with the derived equations. They discover that a minimum magnetic load is necessary to obtain transport within their chip and characterize the transport properties at different magnetic field and loading mass conditions. Overall, the conclusions drawn are reasonable, the figures are well presented, and the paper is well organized. I recommend publication of this manuscript after addressing the following comments:
1. I did not find details about the magnetic beads, such as shape, size, and material properties in the manuscript. This is necessary information to be included.
2. Does changing the density of magnetic particles change the threshold and speed of transportation? It appears to me that the experiments were conducted at a single particle density.
3. The authors show vertical and horizontal transport of the droplet. Is it possible via some activation pattern to obtain diagonal or curved trajectories in this system?
4. There appears to be a mistake in Fig 1b caption, and it should read “… and coil #1 is turned off”.
5. Also consider removing the background audio in the Supplementary movie.
